# Optimization of the Spinneret Rotation Speed and Airflow Parameters for the Nozzleless Forcespinning of a Polymer Solution

**DOI:** 10.3390/polym14051042

**Published:** 2022-03-05

**Authors:** Josef Skrivanek, Pavel Holec, Ondrej Batka, Martin Bilek, Pavel Pokorny

**Affiliations:** 1Department of Textile Machine Design, Faculty of Mechanical Engineering, Technical University of Liberec, 461 17 Liberec, Czech Republic; ondrej.batka@tul.cz (O.B.); martin.bilek@tul.cz (M.B.); 2Department of Nonwovens and Nanofibrous Materials, Faculty of Textile Engineering, Technical University of Liberec, 461 17 Liberec, Czech Republic; pavel.holec@tul.cz (P.H.); pavel.pokorny@tul.cz (P.P.)

**Keywords:** forcespinning, centrifugal, dosing, spinneret, airflow, nanofiber

## Abstract

This paper addresses the changing of the process parameters of nozzleless centrifugal spinning (forcespinning). The primary aim of this study was to determine the dependence of the final product on the dosing of the polymer, the rotation speed of the spinneret and the airflow in order to determine the extent of the technological applicability of aqueous polyvinyl alcohol (PVA) and its modifications. PVA was chosen because it is a widely used polymeric solution with environmentally friendly properties and good biodegradability. It is used in the health care and food packaging sectors. The nanofibrous layers were produced by means of a mobile handheld spinning device of our own construction. This mobile application of the spinning machine has several limitations compared to stationary laboratory equipment, mainly due to dimensional limitations. The uniqueness of our device lies in the possibility of its actual use outside the laboratory. In addition to improved mobility, another exciting feature is the combination of nozzleless forcespinning and fiber application using airflow. Dosing, the rotation speed of the spinnerets and the targeted and controlled use of air comprise the fundamental technological parameters for many devices that operate on a centrifugal force system. The rotation rate of the spinnerets primarily affects the production of fibers and their quality, while the airflow acts as a fiber transport and drying medium. The quality of the fibers was evaluated following the preparation of a testing set for the fiber layers. The most suitable combinations of rotation speed and airflow were then used in subsequent experiments to determine the ideal settings for the device. The solution was then modified by reducing the concentration to 16% and adding a surfactant, thus leading to a reduction in the diameters of the resulting fibers. The nanofiber layers so produced were examined using a scanning electron microscope (SEM) in order to analyze the number of defects and to statistically evaluate the fiber diameters.

## 1. Introduction

Nanofibers are among a number of materials that have recently received considerable attention in the fields of both basic and applied research. They enjoy a huge application potential, and their range of use is constantly expanding, especially in the fields of protective textiles [1], tissue engineering [2,3,4], medicine [5], the production of multispecies materials, energy [6,7], air filtration [8,9,10], contaminated water [11,12] and other substances [13], analytical chemistry [14], etc. [15]. Nanomaterials are also gaining popularity in the field of bioresource fibers, such as the production of nanofibrils [16]. A wide range of manufacturing processes are used for the production of nanofibers, the best-known of which include electrospinning via direct current (DC) or alternating current (AC), drawing [17], forcespinning [18], in exceptional cases melt-blowing [19,20] or other less common methods like mechanotropic spinning [21], and a combination of electrospinning and forcespinning [22,23,24,25]. All these methods, however, feature significant limitations from various perspectives, including low productivity, process complexity or low process safety levels. Hybrid spinning methods are emerging, which combine for example, forcespinning and electrospinning or electrospinning and melt-blowing [26].

Electrospinning comprises one of the most well-known and frequently applied methods for the production of nanofibers. This production method is of particular interest in the scientific sphere from the point of view of the influence of physical fundamentals on the electrostatic process driven by DC [27,28,29] and AC [30,31] nanofiber production techniques. The use of high voltage for the production of nanofibers involves a number of disadvantages, the most important of which concern the dangers of using high voltage, as well as low productivity and the narrow range of materials that can be used for production purposes [32,33,34]. Forcespinning provides an alternative technology for the production of low-diameter fibers. Its main advantages concern the absence of the need for high voltage to form an electric field, a higher production rate and lower energy consumption compared to electrospinning [32], all of which provide sound economic and ecological reasons for the use of this technology for industrial-scale production. 

The physical nature of forcespinning involves the effect of centrifugal forces on the solution or melt that exits the nozzle or the free surface [26]; fibers are formed as a result of the application of force. The resulting fibers are captured on a so-called collector (the part of the spinning device that is used to capture and store the fibers immediately following their formation). The trajectory of the fibers can theoretically be considered spiral [35] (Figure 1). The movement of the fibers can be modified by the application of an airflow that serves to provide a perpendicular direction for the movement of the fibers to the plane in which they are formed; the resulting fiber trajectory can be considered a space helix. The spinning methods that use centrifugal forces are divided into the nozzle (Figure 1a) and nozzle-free (Figure 1b) techniques, depending on whether the liquid emerges from the nozzle or is spun from the surface of the solution. The diameters of the fibers produced via this technology vary between hundreds of nanometers and tens of micrometers.

Figure 1a illustrates the nozzle centrifugal spinning technique. It is a user-safe method, but has a low energy efficiency compared to mass-production polymer spinning approaches. Provided an appropriate polymer solution is used, such as PVA in water, the centrifugal spinning method is ecologically friendly [36,37,38] and safe [[39],[40]. The diameters of nanofibers produced from identical PVA solutions vary according to the production technology used. Fine nanofibers are produced via DC electrospinning, relatively thicker nanofibers are produced via AC technology and forcespinning produces a combination of nanofibers and microfibers [41,42,43,44].

The nozzle to which the spinning solution is supplied rotates at an angular velocity of Ω around its axis of rotation. The force acting on the solution emerging from the nozzle forms a fiber that follows the illustrated trajectory, that is, a theoretical spiral. The fibers are deposited tangentially upon the collector, following which they can be technologically processed. Figure 1b illustrates a similar case; however, the difference lies in the system applied, that is, the presence of so-called fingers instead of a nozzle, which form after overcoming the Rayleigh–Taylor instability state [26,45]. Fibers begin to form after reaching the free edge, that is, the state in which the liquid passes into free space. The movement of fibers to the collector along a spiral trajectory is similar to the case of Figure 1a. The main parameters that affect the fibers themselves and the final formation thereof comprise the viscosity, the surface tension of the used solution, the temperature, the evaporation rate, the centrifugal speed of the spinneret and the distance between the fiber and the collector [41,46,47].

## 2. Materials and Methods

### 2.1. Materials and Solution Preparation

Polyvinyl alcohol dissolved in water was used as the polymeric solution for the ecologically friendly production of nanofibrous materials primarily for future medical purposes [48,49,50] or as a separation membrane component [50,51,52]. PVA solution (20 wt.%) was prepared from PVA purchased from Sigma Aldrich (St. Louis, MI, USA), (M_w_ 31,000–50,000, 98–99% hydrolyzed, Czech Republic) and deionized water. The solution was stirred for 5 h at 80 ℃. A 20 wt.% sodium dodecyl sulfate (SDS) solution (for the modification of the basic PVA solution) was prepared from SDS purchased from Sigma Aldrich (98.5%, Czech Republic) in distilled water [53]. The addition of the surfactant solution was approximately 0.03 g per 5 g of polymeric solution.

### 2.2. Forcespinning Device and Spinning Parameters

A mobile handheld spinning device [54] was assembled as illustrated in Figure 2. A motor-driven spinneret was positioned in the air duct in which centrifugal forces form the fibers. The dosing device delivered a constant 5 mL·min^−1^ of polymeric solution to the spinneret. The airflow generated by the ventilator acted on the resulting fibers and transported them to the collector, where they were applied to a nonwoven fabric. The outer diameter of the spinneret was 30 mm. The distance between the mouth of the channel and the collector with the nonwoven fabric (as the base component for the application of the fibers) was kept constant for all the measurement modifications (i.e., at 320 mm). The inner diameter of the air duct was 103 mm. All forcespinning tests were performed at a temperature of 23 °C and relative humidity of approximately 40%, although humidity changes of about ±5% could occur.

In general, this mobile spinning device is capable of coating various objects to provide protection against a number of influences. The advantage of the applied layer lies in its breathability and filtering ability; therefore, it is essential to monitor the quality of the layer and the parameters that affect its applicability.

A 20% PVA solution was used for the initial experiments. The polymer dosing rate was set at a constant 0.9 mL·min^−1^. Rotation speed in the range 2000–12,000 rpm and flow velocity in the channel in the range of 7.9–11 m·s^−1^ were determined as the settings for the spinning device.

### 2.3. Scanning Electron Microscopy

The microscopic structure of the fibers was observed using a TESCAN Vega 3 scanning electron microscope (TESCAN, Kohoutovice, Czech Republic) at an accelerated voltage of 20 kV with a secondary electron detector. The samples were pre-coated with a 10 nm layer of gold. The histogram fiber diameters were determined using Image-J software from 300 values for each sample.

### 2.4. Viscosity and Surface Tension Measurements

The viscosity of the prepared solutions was determined by a HAAKE Rotovisco rotational viscometer with a C35/1° Ti cone (of angle 1 ℃). The shear rate was linearly increased from 50 to 500 s^−1^. Every sample was measured three times. The surface tension was determined by the bubble pressure tensiometer PocketDyne (Krüss, Hamburg, Germany). All samples were measured ten times in total. For both methods, the obtained data were averaged.

## 3. Results and Discussion

### 3.1. Determination of the Production Potential of the Equipment and the Solution Properties

Measurements were taken aimed at mapping the production potential of the centrifugal spinning device. Table 1 provides a summary of the initial investigation into the dependence of the quality of the fibrous structure on the settings of the spinning device. Spinner speed and channel velocity values outside the above range were also subjected to measurement; however, due to the failure to produce a sufficiently high-quality fiber product, the data obtained was not included in Table 1. The presence of beads, fiber meshes, fiber branching and ribbons was monitored for all the samples.

The measurement of fiber diameters from only a small number of fibers served to provide an initial estimate. The areas selected for subsequent research are highlighted in gray in Table 1. The results indicated that the highest-quality fibrous layers were formed at rotation speed intervals of 5200–8200 rpm and flow velocities in the channel of 7.9–10.1 m·s^−1^. Four types of defects were monitored for the purpose of determining the quality of the fibrous structures: beads, fiber meshes, branching and ribboning. 

The defects that occurred are shown in the images in Figure 3. The most common types of defects comprised beads and fiber meshes, which were observed in various concentrations in practically all the monitored samples. Fiber branching occurred only in those samples produced at excessive spinneret speeds (above 10,500 rpm). The ribboning defect was observed in only one sample, which was most probably a random event that was not directly related to the setup of the device.

Based on the results shown in Table 1, histograms of the fiber diameters were created for selected device setting combinations (Figure 4). The differences in the fiber diameters between the samples at spinneret/flow velocity settings of 5200 rpm/7.9 m·s^−1^ and 5200 rpm/9.2 m·s^−1^ were minimal and ranged from 0.5 to 2.5 μm. The most commonly represented fibers evinced diameters of 1–1.4 μm. The range of diameters for the 8200 rpm/7.9 m·s^−1^ sample was identical to the previously mentioned 0.5–2.5 μm, with fiber diameters of 1.3–1.6 μm. The lowest diameters and the narrowest distributions were attained at a setting of 8200 rpm/9.2 m·s^−1^. The diameters of most fibers ranged between 0.4 and 2 μm, with most below 1 μm. Therefore, the 8200 rpm/10.3 m·s^−1^ setting was used for subsequent measurement purposes. The value of the airflow in the duct was increased from 9.2 rpm to 10.3 rpm in order to avoid the adhesion of fibers to the outer shell of the duct.

Figure 5a shows the prepared solutions’ dynamic viscosities and surface tensions. Although the shear rate increased in the range from 50 to 500 s^−1^, the presented data shows a smaller range of 200–500 s^−1^. The error of the offset of the curve was suppressed in this way. As expected, the 20% PVA solution’s viscosity was higher than that of the 16% PVA solution. Pseudoplastic behavior in adjusted conditions was observed for all solutions. However, the decrease of dynamic viscosity with an increasing shear rate for 16% PVA was insignificant compared to that observed for 20% PVA. In addition, the differences between samples with and without SDS was minor. Figure 5b shows that the addition of SDS to the PVA solution led to a significant decrease in the solution’s surface tension. This subsequently allowed the creation of lower-diameter fibers, as described in the text below.

### 3.2. Production of Fibrous Structures

An optimal setting of 8200 rpm/9.2 m·s^−1^ was determined based on previous findings (see Section 3.1) about the appropriate settings of the spinning device. This setting was applied for the preparation of fibrous layers of 16% and 20% PVA modified via the addition of a surfactant—a 20% aqueous SDS solution. Microscopic examination showed that the layers obtained contained only a small number of droplet defects, while other defects were practically absent, as shown in Figure 6. In both cases, the addition of SDS resulted in a slight reduction in the fiber diameters. Histograms of the fiber diameters confirm this fact (Figure 7).

The variance in the fiber diameters obtained from the primary 16% PVA solution prepared at a setting of 8200 rpm/10.3 m·s^−1^ was approximately 0.3–1.2 μm, with a maximum of around 0.6–0.8 μm. Following the addition of SDS, the maximum shifted to 0.4–0.7 μm. The diameter variance remained the same as for the primary 16% solution. The dispersion of the fiber diameters produced from the 20% solution was between 0.4 and around 2 μm; the most numerous diameters were observed in the range 0.8–1.4 μm. Following the addition of SDS, the scattering of the fiber diameters was reduced to between 0.4 and 1.3 μm, with a shift in the maximum to values of 0.6–0.8 μm. In both cases, the surfactant led to a shift in the frequencies of the fiber diameters to lower values. The SDS lowered the surface tension of the spun solution, thus allowing the creation of thinner fibers.

## 4. Conclusions

The influence of the rotation speed of the spinneret and the airflow on the forcespinning parameters of the handheld spinning device was analyzed. It was determined that these parameters exert a significant influence not only on the manufacturability but, more importantly, on the quality of the resulting blend of nano- and microfiber layers. The value of the centrifugal force acting on the solution affects the formation of fibers. This force affects both the resulting fiber diameter and the occurrence of defects such as beads, fiber meshes, branches and ribbons. The available speed range of the spinner was 2000–12,000 rpm, and the flow velocity in the channel was 7.9–11 m s^−1^.

It was shown that the spinneret speed had a more significant impact on the quality of the final fiber product. Increasing the spinneret rotation speed above 8200 rpm led to fibers of greater diameters. It also slightly increased the number of defects in the fibrous layer. The most frequent defect was fiber mesh, but the proper setting of the spinning device could minimize its appearance. The airflow setting could be modified to optimize the spinning process in fiber transport to the collector and hinder the defiling of the spinning area by overflowing polymer solution. Therefore, the influence of the airflow velocity in the duct was significant mainly from the technological point of view. At low flow rates, the fibers adhered to the inner surfaces of the channel and fluid resistances were formed, which led to variable flow. This resulted in the gradual clogging of the channel and a reduction in the distance required between the formation of fibers and the inner surface of the air duct. Based on the results of the detailed analysis, it was found that the highest-quality fibrous layers were formed after setting the device at a rotation speed of 5200–8200 rpm and a flow velocity in the channel of 7.9–10.1 m·s^−1^. The optimal spinning setting was not influenced by the concentration of the used polymer solution. The forcespinning device was able to spin 16% and 20% PVA solution in almost equal product quality. The final fiber layers showed similar properties, indicating that the presented forcespinning method is not as considerably influenced by the entering polymeric solution as the more common electrospinning process. It was also shown that the modification of the polyvinyl alcohol solution with sodium dodecyl sulfate served to enhance the quality of the produced fibrous layer without the need to change the settings of the spinning machine. Although the used device could not prepare as fine fibers as electrospinning devices, it could be more robust in spun material and work surroundings, making it acceptable for combined spinning technologies.

The used forcespinning device spun layers of micro- and nanofibers of standard quality. The diameters of fibers were 0.7–1.5 μm for 20% PVA and 0.5–1.1 μm for 16% PVA solutions. When compared to other studies, standard forcespinning devices use significantly less-concentrated PVA solutions (6 to 12 wt.%) to prepare fiber layers of similar quality [46,47]. The spinning devices used in previously mentioned studies showed technological limitations to applicable concentrations of the spun polymeric solution, which was not an issue for the instrument used in this paper. Although the presented method cannot fabricate such fine fibers as standard electrospinning technology, which usually range between 0.2 and 0.6 μm for PVA in optimal cases [55], the DC electrospinning technique is limited not only by the concentration of the polymeric solution but also by the geometry of electrodes and presence of HV. Diameters rapidly increase in fibers electrospun from more concentrated solutions than the optimal [56] and reach similar values to the samples presented in this paper. From this point of view, standard DC electrospinning and presented forcespinning should not be considered competing technologies, but more like complementary techniques which could be successfully used together in specific cases. The absence of high voltage, the robustness of the forcespinning method and the mobility of the presented device could have value for other researchers and industries.

Further research will be conducted on the identification of medical and food applications, concerning which nanofibers have a very wide range of potential uses. Moreover, the use of mixtures of nanofibers and microfibers has particular potential in these fields. Microfibers act to improve the mechanical resistance of the product which, in many cases, is beneficial due to the limited mechanical parameters of pure nanofiber layers, which tend to break very easily, thus allowing access for bacteria and viruses from the external environment.

## Figures and Tables

**Figure 1 polymers-14-01042-f001:**
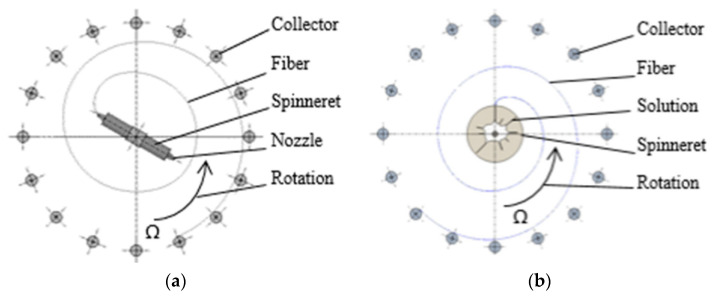
Nozzle centrifugal spinning (**a**) and nozzleless centrifugal spinning (**b**). Angular velocity around the nozzle axis of rotation is represented by Ω.

**Figure 2 polymers-14-01042-f002:**
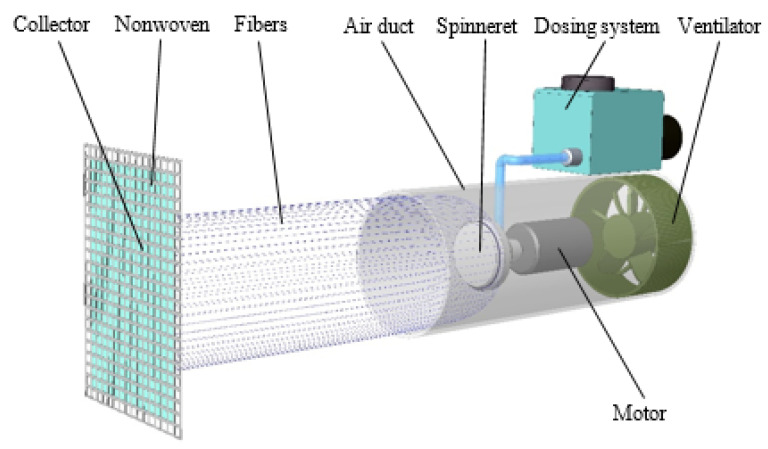
Scheme of the forcespinning system.

**Figure 3 polymers-14-01042-f003:**
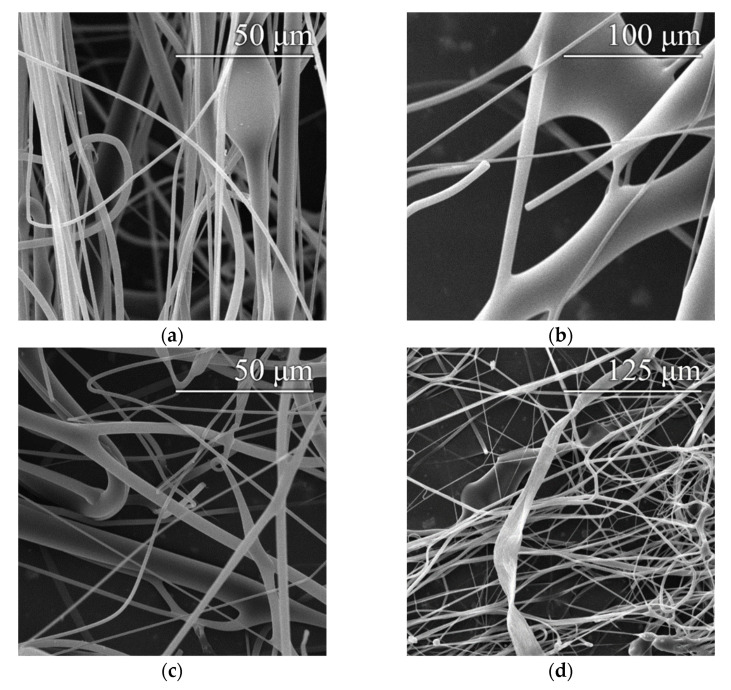
Examples of the observed fiber defects: beads (**a**), fiber mesh (**b**), branching (**c**) and ribboning (**d**).

**Figure 4 polymers-14-01042-f004:**
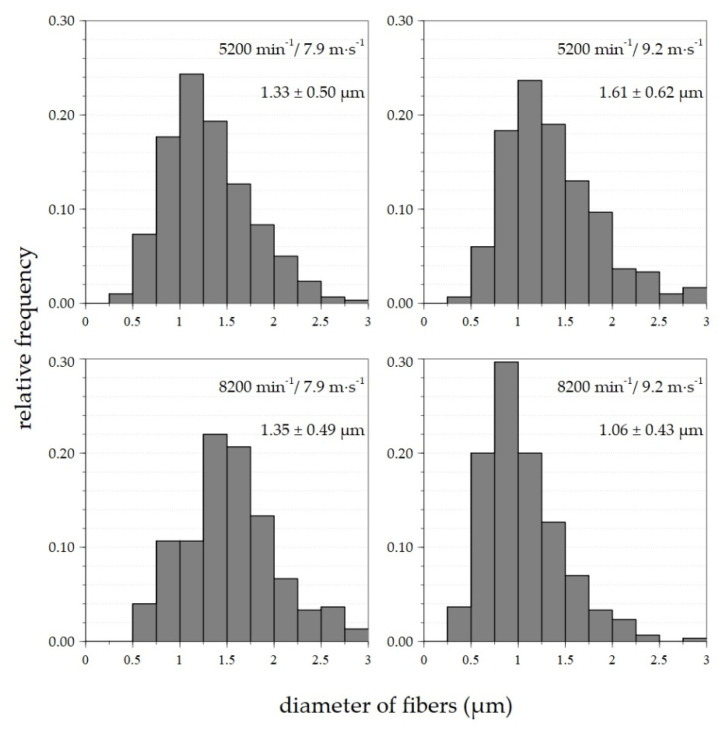
Histograms of fiber diameters with average values and corresponding standard deviations of nanofiber materials made from 20% PVA solution for selected device setting combinations.

**Figure 5 polymers-14-01042-f005:**
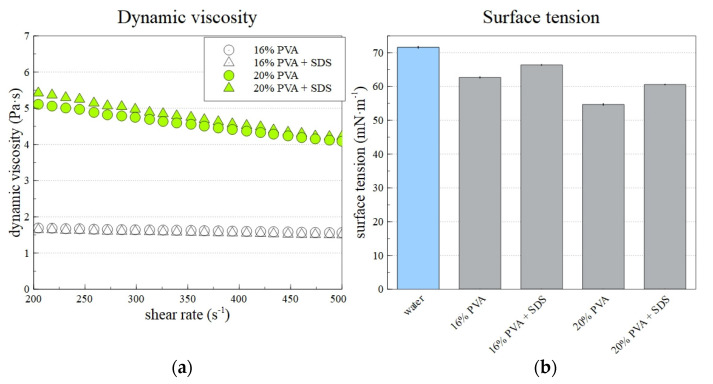
The dynamic viscosity (**a**) and surface tension (**b**) of prepared PVA solutions. The surface tension of distilled water is included to allow better comparison with the PVA samples.

**Figure 6 polymers-14-01042-f006:**
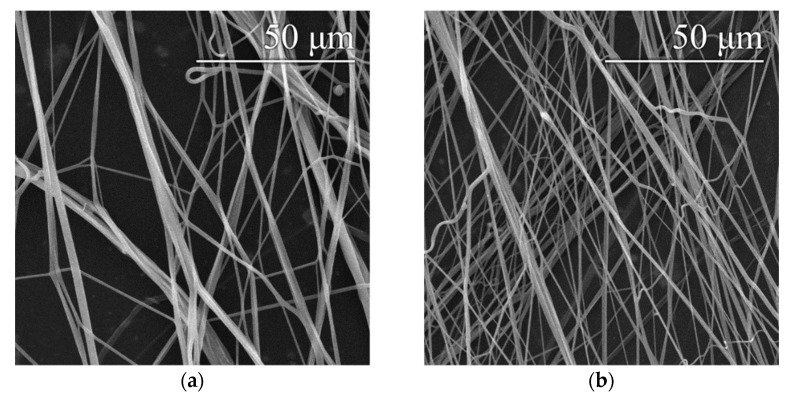
Images of the threads produced with device settings of 8200 rpm/10.3 m·s^−1^–16% PVA (**a**), 16% PVA with SDS (**b**), 20% PVA (**c**) and 20% PVA with SDS (**d**).

**Figure 7 polymers-14-01042-f007:**
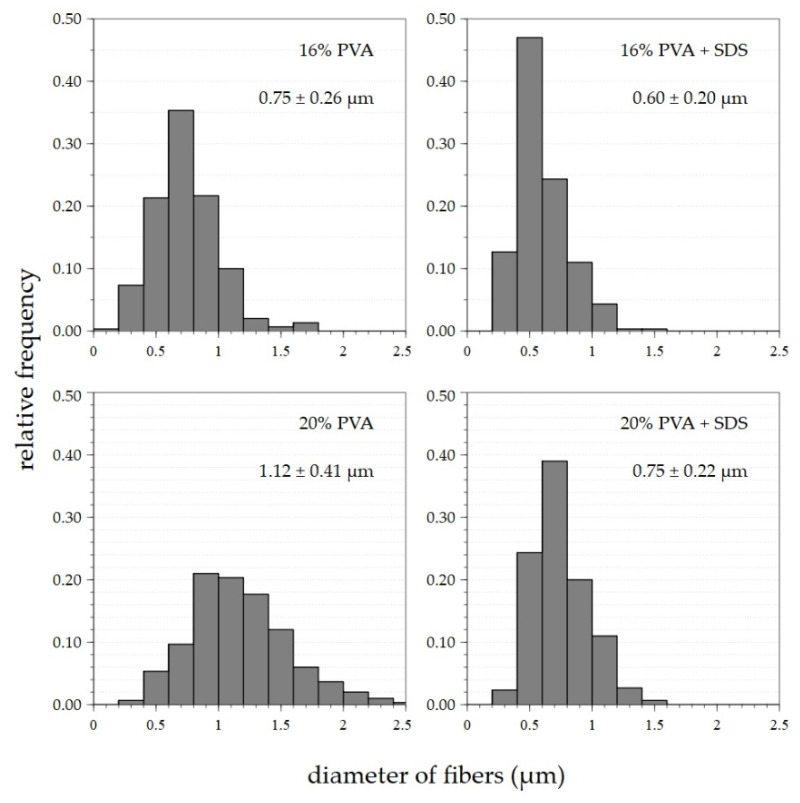
Histograms of fiber diameters with average values and corresponding standard deviations of nanofiber materials prepared with settings of up to 8200 rpm/10.3 m·s^−1^ for the various concentrations of PVA and SDS solutions.

**Table 1 polymers-14-01042-t001:** The dependence of the presence of defects and approximate fiber diameters on spinneret rotation and airflow speed.

Spinneret Rotation (rpm)	Airflow Speed (m·s^−1^)	Beads	Fiber Mesh	Branches	Ribbons	Fiber Diameter (μm)
2000	7.9	yes	-	-	-	2
	9.2	yes	-	-	-	2
	10.3	yes	-	-	-	2
	11	yes	yes	-	-	1.5
5200	7.9	yes	-	-	-	1.5
	9.2	-	yes	-	-	1.5
	10.3	-	yes	-	-	2
	11	yes	yes	-	yes	2.5
8200	7.9	yes	yes	-	-	1.5
	9.2	-	yes	-	-	1.5
	10.3	yes	-	-	-	1.5
	11	yes	-	-	-	1
10,000	7.9	yes	yes	-	-	2.5
	9.2	yes	-	-	-	2
	10.3	yes	-	-	-	2.5
	11	-	yes	-	-	2.5
12,000	7.9	yes	-	-	-	3
	9.2	yes	-	yes	-	3.5
	10.3	yes	yes	yes	-	3
	11	yes	-	-	-	2.5

## Data Availability

The data presented in this study is available on request from the corresponding author.

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
