# Peer review of "Optimization of the Spinneret Rotation Speed and Airflow Parameters for the Nozzleless Forcespinning of a Polymer Solution"

_polymers, 2022, doi:10.3390/polym14051042_

Round 1

Reviewer 1 Report

This is a solid work for the optimisation of the spinneret rotation speed and airflow parameters for the nozzleless forcespinning of a polymer solution. It is mainly focused on validation of the methodology and could be of interest to researchers in the area of electrospun polymer. I just have a few minor suggestions before I can recommend it for publication.

(1) The authors may consider to explore the mechanical property of the fibers obtained as well.

(2) In the introduction, the authors give an overview of current development and applications of fibers. However, the authors may have missed a few papers based on bioresource fibers, such as DOI: 10.1016/j.jobab.2020.03.002; DOI: 10.1016/j.jobab.2020.03.003. Those new development should be incorporated into the overview since they represent a new trend in the area.

Based on the above concerns, I suggest a minor revision.

Author Response

Dear Reviewer 1,

Colleagues and I appreciate the revision of our paper. We have corrected the paper the text according to your suggestions in the following reply.

Sincerely,

Josef SkÅ™ivánek

  • The authors may consider to explore the mechanical property of the fibers obtained as well.

Our response:          It is a perfect idea for follow-up research. Sophisticated test equipment (INSTRON 5943, USA) could be used to examine mechanical parameters. However, it is generally recommended to determine these parameters as soon as possible after fiber production to avoid ageing and degradation of the fibers over time. Other determination of mechanical parameters would be wildly inaccurate due to the time lag from the production of the examined fibers in this work.

  • In the introduction, the authors give an overview of current development and applications of fibers. However, the authors may have missed a few papers based on bioresource fibers, such as DOI: 10.1016/j.jobab.2020.03.002; DOI: 10.1016/j.jobab.2020.03.003. Those new development should be incorporated into the overview since they represent a new trend in the area.

Our response:          The recommended paper Utilization of discarded crop straw to produce cellulose nanofibrils and their assemblies has been added to the introduction (lines 40-41).

Reviewer 2 Report

After reading the author's answers to my previous comments, I don't feel they make an effort to attend to such comments. Hence, after asking to improve their discussion comparing their results, they just mentioned that is difficult to find other studies with their parameters. I think that the idea is not to find an exact paper with the same parameter but to compare other studies that used the force spinning to create fibers with PVA (or another polymer), and what they obtained compared to the authors finding, what they do different and what improvement they got, compared to the reported results.

Here there are some papers that's fabricated PVA fibers with forcespinning

https://onlinelibrary.wiley.com/doi/epdf/10.1002/adv.21817

https://doi.org/10.4028/www.scientific.net/AMR.1123.20

https://fr.art1lib.org/book/45972447/157075

You can also compare your results with PVA electrospun fibers. The idea is to compare results, more extensively, explaining, justifying, and proposing ideas to improve your results. In this actually improved paper, no references are added in the results and discussion section, so no interesting arguments are seen 

Also, in another response, I ask if the absence of HV is the only interesting parameter to discuss, and the author's response was that there are other ways to compare, but no further arguments are given

The abstract is not well structured, no justification of the importance of the study is described

Still the argument novelty "study the relation between forcespinning setting and produced fibers" is not a justified novelty for this journal, because without making any experiments is obvious that the rotation speed of the spinneret and the airflow on the forcespinning parameters will affect the resultant fibers. Maybe if you the authors proposed this work as a technical report in a more adequate journal will be a successful publication.

Moreover, I don't feel this study fits with the scope of the journal which is:

Polymers provides an interdisciplinary forum for publishing papers which advance the fields of (i) polymerization methods, (ii) theory, simulation, and modeling, (iii) understanding of new physical phenomena, (iv) advances in characterization techniques, and (v) harnessing of self-assembly and biological strategies for producing complex multifunctional structures

I don't feel that meanwhile the article is not argumented properly, is not suitable for publication in this important journal

Author Response

Dear Reviewer 2,

Colleagues and I appreciate the revision of our paper. We have corrected the paper according to your suggestions in the following reply.

Sincerely,

Josef SkÅ™ivánek

  • After reading the author's answers to my previous comments, I don't feel they make an effort to attend to such comments. Hence, after asking to improve their discussion comparing their results, they just mentioned that is difficult to find other studies with their parameters. I think that the idea is not to find an exact paper with the same parameter but to compare other studies that used the force spinning to create fibers with PVA (or another polymer), and what they obtained compared to the authors finding, what they do different and what improvement they got, compared to the reported results.

Our response:      The discussion comparing our results to the studies that have fabricated PVA fiber with forcespinning was added (lines 237-250). Thank you for recommended studies that are interesting to compare with our work.

  • Comparing our results with PVA electrospun fibers.

Our response:      The conclusion now presents a comparison between the fibers from the PVA solution obtained by electrospinning and forcespinning.

  • Absence of HV:

Our response:      Lines 51-53 show other advantages of forcespinning compared to the most common method of nanofiber production, which is electrospinning.

  • The abstract is not well structured, no justification of the importance of the study is described

Our response:      Thank you for pointing out the importance of our work (lines 17-21).

  • Still the argument novelty ….. the rotation speed of the spinneret and the airflow on the forcespinning parameters will affect the resultant fibers.

Our response:      It is obvious that these parameters affect the resulting fibers, but our goal was to find out how their dimensions and structure are affected by the given technological conditions. We decided to publish this article in this special issue, as we were advised to by the Guest Editor's (in cc).

  • I don't feel that meanwhile the article is not argumented properly, is not suitable for publication in this important journal

Our response:      This study can be included in this journal. Forcepinnig is another interesting alternative to electrospinning. The paper can be an inspiration for expanding research directions. There are applications where forcespinnig is combined as an accompanying method to electrospinning. (in lines 50)

Reviewer 3 Report

This is an interesting work devoted to the method of high-performance production of non-woven materials with micron fibers without coagulation baths. The subject of discussion, the method, and the spinning device are interesting and relevant. Recently, many publications have been made on a similar theme. There are some questions about the experiments, the answers to which will strengthen the manuscript.

1) Line 50. In addition to electrospinning and centrifugal spinning, there is a mechanotropic process (doi: 10.3390/polym10080856 ) in which fiber formation occurs due to spontaneous thinning of a viscoelastic solution under the action of capillary, inertial, and viscoelastic forces (doi: 10.1063/5.0060960 ). That technology makes it possible to obtain a continuous fiber without the use of coagulation baths.

2) Line 84 A huge number of world-famous rheologists deal with the jet instabilities problems of viscoelastic solutions under the action of various forces. I think it would be good to refer to works that deal with different types of instabilities. Highly recommend one of the latest reviews on this topic where you can find that publications (doi: 10.1070/RCR4941)

3) Unfortunately, it was unable to access the article [43] because it is not available in major publishers MDPI, Elzivir, Taylor, Springer, etc. As I understand it, you have Nozzle-less centrifugal spinning. What is the spinneret configuration used and what is the gap through which the solution flows?

4) Line 124-125 Why did you use a linear scale? Logarithmic dependences are usually taken in rheological measurements, according to which the properties of solutions are visible.

5) Line 133-135. The experiment parameters should be moved to the Experimental section (part 2.1).

6) Line 142. Has it been found any parameters for defectless spinning?

7) Line 146. Why not to 10000 rpm following Table 1?

8) Line 152. Table 1 notes that fiber mesh is formed at lower speeds (since 2000)

9) Line 172. It would be preferable to see the full flow curve in traditional log(η) / log (ɣ̇) coordinates. Perhaps this will allow you to more clearly see the difference between the solutions behavior. And of course, the viscoelastic properties of solutions are more interesting.

10) Line 179 Here are very strange data on surface tension, they are not combined with literature data. Typically, a polymer addition lowers the surface tension. This issue is also well studied for PVA-water solutions. Here are some examples:

doi: 10.3390/molecules25204799 (Fig 4a)

doi: 10.1002/app.20436

doi: 10.1021/acs.jpcb.8b08374

11) Line 188 The description of the amount of SDS added should be moved to section 2.2.

12) Line 211 The conclusions should be expanded. What parameters improve fiber formation? What is the difference between the two concentrations? Why exactly concentrations were chosen?

Minor flaws:

13) Line 114 There is space between number and degree symbol should be.

14) Lines 102, 103, 116 A number and “%” should be written without a space symbol.

15) Line 124. In this case, this is not a very important point. However, if you indicate a cone, then it is worth specifying the angle of the cone for the reader to understand reliable measurement intervals. At the same time, the gap is not important at al. Because it is determined by the cone geometry and is always strictly defined to establish a constant shear rate from the center to the edge of the measuring unit.

16) Line 135, etc. 200012000 should be written

Author Response

Dear Reviewer 3,

Colleagues and I appreciate the revision of our paper. We have corrected the paper according to your suggestions in the following reply.

Sincerely,

Josef SkÅ™ivánek

  • Line 50. In addition to electrospinning and centrifugal spinning, there is a mechanotropic process (doi: 10.3390/polym10080856 ) in which fiber formation occurs due to spontaneous thinning of a viscoelastic solution under the action of capillary, inertial, and viscoelastic forces (doi: 10.1063/5.0060960 ). That technology makes it possible to obtain a continuous fiber without the use of coagulation baths.

Our response:     The mechanotropic spinning was mentioned in the text. In order to keep the topic of the paper coherent only first reference was used, because it deals directly with previously not-mentioned spinning technology, whereas the second suggested article add rather deeper insight into the mechanotropic method of fiber creation.

  • Line 84 A huge number of world-famous rheologists deal with the jet instabilities problems of viscoelastic solutions under the action of various forces. I think it would be good to refer to works that deal with different types of instabilities. Highly recommend one of the latest reviews on this topic where you can find that publications (doi: 10.1070/RCR4941)

Our response:     The viscoelastic behaviour of the used solutions greatly influences the spinning process. However the area of interest of our paper deals with the actual process and the prepared fibers more than with the theoretical background of forcespinning.
We added one reference as suggested to the Tylor’s cone issue but we believe that it should be enough for the basic ability to understand our paper’s topic.

  • Unfortunately, it was unable to access the article [43] because it is not available in major publishers MDPI, Elzivir, Taylor, Springer, etc. As I understand it, you have Nozzle-less centrifugal spinning. What is the spinneret configuration used and what is the gap through which the solution flows?

Our response:     The reference [43] leads to the Industrial Property Office of Czech Republic and the corresponding Utility model of the used spinning machine. We checked the attached link and it should lead directly to the Utility model. We believe that the reference supports the usability of the machine and shows that it is not only a self-made laboratory apparatus.

The spinning system uses a free outlet from the dosing tube to the inner surface of the rotating cone of the spinneret. By the action of the centrifugal force, a liquid film is formed, which moves in the direction of the increasing diameter of the cone. As soon as the moving liquid film reaches the largest diameter at the free end of the spinneret, fibers begin to form due to the effects of centrifugal forces. This system is very robust, as there is no clogging of the fluid ways, due to the absence of nozzles and gaps. On the other hand, the system is demanding on accurate dosing. The described design solution is protected by utility model No. 30004 and some authors of this article are its originators, see [43].

  • Line 124-125 Why did you use a linear scale? Logarithmic dependences are usually taken in rheological measurements, according to which the properties of solutions are visible.

Our response:     We agree that usually logarithmic dependences are used, however our claim was to show the difference between used polymeric solutions in the simplest way. Due to this only a limited intervals of shear rates were tested and it was confirmed that the more concentrated solutions were more viscous (the addition of SDS almost did not change the viscosity). If this short interval would be displayed in logarithmic scale, it shows very similar curves thus the information would be the same. Therefore, the simpler linear axes were chosen for the presented graph.

  • Line 133-135. The experiment parameters should be moved to the Experimental section (part 2.1).

Our response:     The experimental parameters were moved to section 2.1 as suggested. Additionally, minor improvements of the text were realized to better fit the text to the mentioned section. The Material and solution preparation section was also moved up, because it presents used solutions, to which the original text referred to. This should help the chapter to be more consecutive.

  • Line 142. Has it been found any parameters for defectless spinning?

Our response:     The found parameters for defectless or the least defectless spinning are shown in Table 1 and they are highlighted by grey colour. The more detailed examination of this topic is presented in the 3.2 section.

  • Line 146. Why not to 10000 rpm following Table 1?

Our response:     10000 rpm was not chosen for further examination because the diameters of prepared fibers were excessively high (more than the diameter from 5200 and 8200 rpm setting). Table 1 shows that rising rotation speed (above cca 8200 rpm) lead to creation of thicker fibers. Due to this we estimated that 5200 and 8200 rpm would be more relevant for subsequent experiments.

  • Line 152. Table 1 notes that fiber mesh is formed at lower speeds (since 2000)

Our response:     We stated that fiber mesh defects were observed in almost all rpm sample sets (lines 150-152).

  • Line 172. It would be preferable to see the full flow curve in traditional log(η) / log (ɣ̇) coordinates. Perhaps this will allow you to more clearly see the difference between the solutions behaviour. And of course, the viscoelastic properties of solutions are more interesting.

Our response:     The whole flow curve was not presented, due to the pseudoplastic behaviour of our solutions. After reaching certain shear rate values the dynamic viscosities were of the similar values for all four solutions. Therefore, we chose to show only part of the curves, where the differences between samples were distinct. The log/log coordinates were tested for the graph, but the information value of the resulting curves was almost the same as for the simple linear coordinates. Hence the simpler way was chosen for the presented graph.

  • Line 179 Here are very strange data on surface tension, they are not combined with literature data. Typically, a polymer addition lowers the surface tension. This issue is also well studied for PVA-water solutions. Here are some examples: doi: 10.3390/molecules25204799 (Fig 4a), doi: 10.1002/app.20436, doi: 10.1021/acs.jpcb.8b08374

Our response:     It is true that previously the stated values are higher than in literature but we aimed to show the decrease of the surface tension after the addition of SDS. We have re-measured the values and figured out, that the initial test was affected by the wrongly adjusted lifespan of the bubbles, which did not affect the values for water but did for the polymeric solutions. The corresponding graph on Figure 5 was modified to show the re-measured values.

  • Line 188 The description of the amount of SDS added should be moved to section 2.2.

Our response:     The modification of polymeric solutions by SDS adition was moved to the suggested section.

  • Line 211 The conclusions should be expanded. What parameters improve fiber formation? What is the difference between the two concentrations? Why exactly concentrations were chosen?

Our response:     The conclusion was rewritten to be more coherent and clear. The tested parameters were commented and the difference between the used solutions was added.
The two concentration of PVA was chosen for examination of the robustness of the spinning method. The difference of 4 % between the solutions should be significant (at least for electrospinning) and the results shown that the forcespinning device was not so much affected by this change of entering polymeric solution. To provide even more variability of the entering material SDS enhanced solutions were tested.

Minor flaws:

Our response:     All minor flaws were corrected as suggested.

  • Line 114 There is space between number and degree symbol should be.
  • Lines 102, 103, 116 A number and “%” should be written without a space symbol.
  • Line 124. In this case, this is not a very important point. However, if you indicate a cone, then it is worth specifying the angle of the cone for the reader to understand reliable measurement intervals. At the same time, the gap is not important at al. Because it is determined by the cone geometry and is always strictly defined to establish a constant shear rate from the center to the edge of the measuring unit.

Our response:     The necessary information about the gap was deleted. The cone angle was added to the text.

  • Line 135, etc. 2000–12000 should be written

Round 2

Reviewer 2 Report

The authors have improved the manuscript giving the enough quality to be published in this important journal

Reviewer 3 Report

The authors have answered the questions, I have no major remarks about the Manuscript.

Minor flaws:

Authors should improve the presentation quality of the manuscript because there are some typos.